# SupGRPO: Enhancing GRPO with Matching-based Online SFT for Text Spotting

## Abstract

Text spotting requires both accurate text recognition and precise spatial localization. Current specialised spotters excel at predicting tight bounding boxes in natural scenes, but falter on complex or artistic text, whereas multimodal large language models (MLLMs) possess strong recognition capabilities yet remain weak at localisation. To equip the text spotter with general and powerful recognition capabilities and to maximize its localization ability, we explore two MLLM-based fine-tuning methods: Supervised Fine-Tuning (SFT) and reinforcement learning fine-tuning based on Group Relative Policy Optimisation (GRPO). An interesting finding is that SFT is less effective than GRPO at enhancing recognition, while GRPO is less effective than SFT at enhancing detection. To compensate for each other's shortcomings, we introduce a joint training strategy, SupGRPO, which simultaneously optimizes the model using both SFT and GRPO. SupGRPO employs the specially designed reward functions and develops a matching-based online SFT applied solely to coordinate tokens. It both mitigates the reward sparsity problem of GRPO and avoids the instance order dependency problem of SFT. To evaluate particularly challenging cases, we curate ATS, a dataset for artistic text spotting. Experiments demonstrate that SupGRPO improves both text recognition and detection, validating the proposed approach. We will release ATS and our code upon acceptance.

## 1 Introduction

Text spotting is a crucial task in computer vision that involves simultaneously detecting the location and recognizing the content of text instances in images. Significant progress has been made by specialized text spotting models on natural scene images, with diverse approaches including segmentation-based methods like Mask TextSpotter (Lyu et al., 2018), regression-based methods utilizing techniques such as Bezier curves in ABCNet (Liu et al., 2020; 2021), and transformer-based models like TESTR (Zhang et al., 2022) and DeepSolo (Ye et al., 2023). However, these methods often struggle substantially when faced with more intricate scenarios such as artistic text. These scenarios frequently exhibit stylized fonts, unconventional layouts, intricate textures, and complex visual integration, demanding advanced visual understanding and reasoning capabilities that often exceed the capacity of models designed for normal scene text.

Concurrently, Multimodal Large Language Models (MLLMs) have demonstrated powerful visual understanding abilities across a wide range of tasks, including advancements in OCR-related areas such as document understanding and scene text analysis, exemplified by models like TextMonkey (Liu et al., 2024), Vary (Wei et al., 2024a), InternVL3.5 (Wang et al., 2025) and Qwen2.5-VL (Bai et al., 2025b). These capabilities suggest that MLLMs hold great potential for addressing the complexities inherent in challenging text scenarios like artistic text. Nevertheless, while these MLLMs show promising performance in text recognition and understanding, their ability to precisely predict the fine-grained location coordinates required for robust text spotting remains notably limited. The text spotting results from different types of models are shown in Fig. 1 (a).

To equip the text spotter with general and powerful recognition capabilities and to maximize its localization ability, we attempt to fine-tune the MLLM using SFT and GRPO separately. According to the exploration in Sec. 4.4, an interesting finding is that SFT is less effective than GRPO at enhancing recognition, while GRPO is less effective than SFT at enhancing detection. First, we adopt

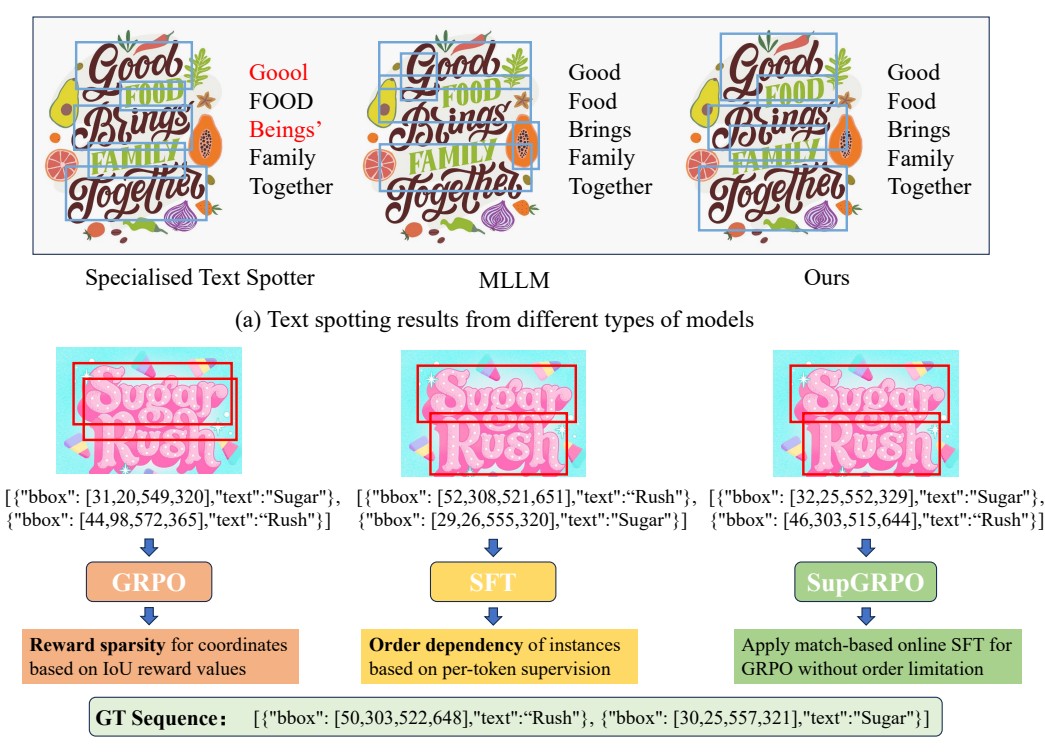

(a) Text spotting results from different types of models

(b) SupGRPO combines GRPO and SFT and avoids their respective problems

Figure 1: Comparison with other models and other training strategies.

the GRPO (Shao et al., 2024) training paradigm and apply it for the first time to the text spotting task. Within this adopted paradigm, we carefully craft rule-based reward functions tailored to guide the model's optimization for text spotting. Specifically, we utilize four rewards: a format reward for structured output, a text content reward for accurate text recognition, an IoU precision reward and an IoU recall reward for accurate text detection. Using GRPO-based reinforcement learning (Shao et al., 2024; Guo et al., 2025) enables the direct optimization of non-differentiable evaluation metrics (IoU and F1-score) and encourages reasoning exploration in ambiguous artistic scenes. However, the sparse reward signal provided by GRPO lacks the fine-grained, direct supervision that is essential for achieving highly accurate position coordinate prediction.

On the other hand, SFT can provide the model with token-level, fine-grained supervision, which is highly effective for the model to acquire accurate positional coordinates. However, for the text spotting task, SFT presents the instance order dependency problem, which often imposes an arbitrary and meaningless order among independent text instances. GRPO can effectively resolve this by optimizing for the set of predictions via order-agnostic rewards. Thus, it is essential to integrate SFT and GRPO within an end-to-end framework to achieve synergistic collaboration. Fig. 1 (b) illustrates the inherent features of different training strategies.

To address the limitations of both GRPO's reward sparsity for coordinates and standard SFT's sequence dependency, we propose **SupGRPO**, building upon the GRPO framework. SupGRPO applies a more targeted, matching-based online SFT specifically to the coordinate tokens of the predicted text instances. Specifically, for each output sequence sampled during GRPO training, we extract the coordinate tokens based on a regularized matching process. Then, each predicted text box derived from these coordinate tokens is matched with a corresponding ground truth (GT) text box. Finally, a per-token loss is calculated based on this matching, providing direct, token-level supervision for localization accuracy. This approach effectively leverages the benefits of SFT for precise coordinate learning while avoiding the drawbacks of standard SFT. Ultimately, we jointly train an MLLM using GRPO and the matching-based online SFT.

Existing scene text spotting datasets often do not adequately represent the artistic and complex scenarios that require strong visual understanding and reasoning, limiting the evaluation of models on such challenging data. In view of this, we construct an artistic text spotting dataset named **ATS** with $6.5k$ training images and $2.5k$ testing images. Experiments show the superiority of SupGRPO in both text detection and end-to-end recognition. Ablation study shows that while GRPO significantly boosts recognition performance, SFT is more effective for improving detection accuracy, verifying the need for a combined approach.

In summary, the main contributions are threefold:

(1) To the best of our knowledge, this work is the first to expand the capability boundaries of MLLMs specifically for the text spotting task. Furthermore, it is the first academic study focused on the challenging scenario of artistic text spotting, moving beyond standard scene text.

(2) We introduce an end-to-end training strategy SupGRPO. This design resolves the instance order dependency of standard SFT and the reward sparsity of vanilla GRPO. We also achieve a synergy where SFT enhances detection precision while GRPO boosts recognition reasoning.

(3) We construct a standardized dataset for artistic text spotting. We perform comprehensive evaluations of both specialized models and MLLMs on this benchmark, establishing a critical reference point for future research in this domain.

## 2 RELATED WORK

### 2.1 TEXT SPOTTING METHODS

Text spotting integrates text detection and recognition for simultaneous processing, exploiting their complementarity. Early methods like SEE (Bartz et al., 2018) and those utilizing varying-size RoI Pooling (Li et al., 2017) focused on horizontal or rotated text, often employing techniques like bilinear sampling (Busta et al., 2017) or RoI-Rotate (Liu et al., 2018) for feature alignment but struggling with curved or artistic text. This led to the development of arbitrarily-shaped text spotting, with significant progress made by segmentation-based methods, such as Mask TextSpotter (Lyu et al., 2018) and its improvements (Liao et al., 2021; 2020), which use mask branches for instance and character segmentation, or methods like PGNet (Wang et al., 2021) which are single-shot and avoid character-level annotations. Concurrently, regression-based approaches achieved advanced performance by focusing on more accurate shape representations and feature sampling. These methods include using differentiable RoISlide (Feng et al., 2019), parameterized Bezier curves with BezierAlign in ABCNet (Liu et al., 2020; 2021), and recent DETR (Carion et al., 2020; Zhu et al., 2020)-based methods like TESTR (Zhang et al., 2022) and DeepSolo (Ye et al., 2023), or sequence-based methods like SPTS (Peng et al., 2022) and UNITS (Kil et al., 2023). ESTextSpotter (Huang et al., 2023) proposes to achieve explicit synergy between text detection and recognition. Bridge (Huang et al., 2024) effectively connects the well-trained detector and recognizer. While these methods greatly advanced the handling of arbitrarily-shaped text, they often struggle with more complex and challenging scenarios, such as artistic text.

### 2.2 TEXT SPOTTING DATASETS

For scene text spotting, there is a synthetic dataset SynthText150K (Liu et al., 2020; 2021) proposed by Liu *et al.* to enrich the arbitrarily shaped scene text for training. Total-Text (Ch'ng et al., 2020) contains 1,255 images for training and 300 images for testing. It is an arbitrarily-shaped scene text benchmark with polygon annotation on the word level. SCUT-CTW1500 (Liu et al., 2019) includes 1,000 training images and 500 testing images. It is also an arbitrarily-shaped scene text benchmark but with text-line level annotation. It contains both English and Chinese text. ICDAR 2015 (Karatzas et al., 2015) provides images captured by Google Glass in the real world, so there are many instances with small sizes, low resolution, or any orientation. It consists of 1,000 training images and 500 testing images. However, these datasets cannot represent complex scenes that rely on strong visual understanding and reasoning abilities.

## 2.3 MULTIMODAL LARGE LANGUAGE MODELS FOR OCR

Recent advancements in multimodal large language models (MLLMs) have significantly enhanced optical character recognition (OCR) capabilities, particularly in document understanding and scene text analysis. Models such as TextMonkey (Liu et al., 2024) achieve promising performance on text VQA and text spotting using shifted window attention and token resampling. TextHawk2 (Yu et al., 2024) enables efficient fine-grained perception with high token compression. Vary (Wei et al., 2024a) expands the vision vocabulary for improved document parsing. Ocean-OCR (Chen et al., 2025) addresses variable resolution inputs, and UniDoc (Feng et al., 2023) focuses on unified multimodal instruction tuning for simultaneous text detection, recognition, spotting and understanding. mPLUG-DocOwl 1.5 (Hu et al., 2024) introduces multi-grained text localization and a novel H-Reducer module to enhance OCR-free document understanding across diverse domains. Some recent general MLLMs such as InternVL3.5 (Wang et al., 2025) and Qwen2.5-VL (Bai et al., 2025b) show better performance in complex document understanding and text detection. Although these MLLMs have shown leading performance in text recognition and document understanding, their ability to accurately predict location coordinates is still very limited.

## 3 METHOD

### 3.1 PRELIMINARY

Group Relative Policy Optimization (GRPO) is an efficient reinforcement learning algorithm initially proposed in the DeepSeekMath model to enhance mathematical reasoning capabilities in large language models (Shao et al., 2024). Unlike standard Proximal Policy Optimization (PPO) (Schulman et al., 2017), GRPO does not estimate a value function. Instead, it directly computes the advantage function based on relative rewards from multiple outputs sampled from the same input, simplifying training and reducing computational overhead.

Specifically, GRPO trains the policy $\pi_\theta(y|x)$ to maximize an expected reward $R(y)$. It operates by sampling a group of $G$ output sequences $\{y_1, \ldots, y_G\}$ for a given input $x$ from the policy. The objective function involves maximizing the expected relative advantage over these sampled outputs:

$$\mathcal{J}_{\text{GRPO}}(\theta) = \mathbb{E}_x \left[ \frac{1}{G} \sum_{i=1}^{G} \frac{1}{|y_i|} \sum_{t=1}^{|y_i|} \left[ \frac{\pi_\theta(y_{i,t}|x, y_{i,<t})}{\pi_{\theta_{\text{old}}}(y_{i,t}|x, y_{i,<t})} \hat{A}_{i,t} \right] - \beta \cdot \text{KL}(\pi_\theta || \pi_{\text{ref}}) \right], \quad (1)$$

where $\pi_{\theta_{\text{old}}}$ is a past policy, $\pi_{\text{ref}}$ is a reference policy, $\beta$ is a coefficient, and $\hat{A}_{i,t}$ is the advantage for token $y_{i,t}$. A core aspect of GRPO is the computation of $\hat{A}_{i,t}$ based on the *relative rewards* within the sampled group, avoiding a separate value function model. The advantage can be based on the normalized reward $\tilde{r}_i$ for the sequence $y_i$:

$$\hat{A}_{i,t} = \tilde{r}_i = \frac{R(y_i) - \text{mean}(R(\{y_j\}_{j=1}^{G}))}{\text{std}(R(\{y_j\}_{j=1}^{G}))}. \quad (2)$$

As presented in DeepSeekMath (Shao et al., 2024), various training methods like SFT and GRPO can be understood under a unified framework for training generative policies. They optimize the same policy but differ in their objectives and how gradients are derived. The gradient can be expressed as:

$$\nabla_\theta J(\theta) = \mathbb{E}_{(x,y)\sim D} \left[ \frac{1}{|y|} \sum_{t=1}^{|y|} GC(x, y, t) \nabla_\theta \log \pi_\theta(y_t|x, y_{<t}) \right]. \quad (3)$$

This highlights three components: 1) *Data Source $D$*; 2) *Reward Signal Source*; and 3) *Gradient Coefficient $GC$*. SFT uses ground truth data ($D = \mathcal{D}_{\text{SFT}}$) with $GC = 1$. GRPO samples from the policy ($D \leftarrow \pi_\theta$) using a reward model and computes $GC$ based on group relative advantages. This framework provides a common lens to view distinct policy optimization strategies. Therefore, the objective function of SFT is:

$$\mathcal{J}_{SFT}(\theta) = \mathbb{E}_x \left( \frac{1}{|y|} \sum_{t=1}^{|y|} \log \pi_\theta(y_t \mid q, y_{<t}) \right). \quad (4)$$

## 3.2 Reward Functions for Text Spotting

To effectively guide the GRPO training towards generating accurate text spotting results, we design four task-specific reward functions. These rewards evaluate different aspects of the model's output, providing a comprehensive signal for policy optimization. The total reward $R(y)$ for a generated output sequence $y$ is a combination of these individual reward components.

**Format Reward** This reward ensures that the generated sequence $y$ adheres to the expected structured format, which is a list of dictionaries, where each dictionary contains a "bbox" key with a list of four numerical coordinates and a "text" key with the recognized text string. If the output sequence conforms to this predefined format, the model receives a reward of 1; otherwise, the reward is 0. This encourages the model to generate outputs that can be parsed for subsequent evaluation.

$$R_{\text{format}}(y) = \begin{cases} 1, & \text{if } y \text{ follows the specified format} \\ 0. & \text{otherwise} \end{cases} \tag{5}$$

**Text Reward** This reward evaluates the accuracy of the recognized text content. We collect all predicted text strings from the parsed output and all ground truth text strings from the annotation. We then treat these as two sets of words and compute the word-level F1 score between them. Let $P_{\text{words}}$ be the set of all words from the predicted text strings in $y$, and $GT_{\text{words}}$ be the set of all words from the ground truth text strings in $GT$. The word-level F1 score is calculated based on the intersection and union of these two collections of words:

$$R_{\text{text}}(y, GT) = F1_{\text{word}} = \frac{2 \cdot |P_{\text{words}} \cap GT_{\text{words}}|}{|P_{\text{words}}| + |GT_{\text{words}}|}, \tag{6}$$

where $|P_{\text{words}} \cap GT_{\text{words}}|$ represents the count of words common to both sets, and $|P_{\text{words}}| + |GT_{\text{words}}|$ represents the total count of words in both sets. Note that our implementation treats predicted and ground truth words as *Multisets (Bag of Words)* to handle duplicate instances. Specifically, the intersection count for a given word $w$ is calculated as $\min(\text{count}(w, P_{words}), \text{count}(w, GT_{words}))$. This mechanism ensures that valid duplicates (e.g., "Sale" appearing twice in the image) must be predicted the correct number of times to achieve full recall. Conversely, if the model hallucinates duplicates (e.g., predicting "Together" twice when it appears only once), the redundant prediction contributes to the total prediction count $|P_{words}|$ but not the intersection, thereby correctly penalizing the precision score.

We use the word-level F1-score to strictly align with the standard evaluation protocols for text spotting benchmarks, which typically rely on exact match criteria. Besides, a fundamental advantage of reinforcement learning is its ability to directly optimize the non-differentiable evaluation metrics. Using *soft* rewards like ANLS or Edit Distance introduces a misalignment: it encourages the model to learn *approximate* spellings (e.g., "Applo" vs. "Apple") to maximize partial rewards, whereas the test protocol penalizes any character error as a complete failure.

**IoU Precision Reward** This reward assesses the overall precision of the detected bounding boxes. We compute the detection precision based on an Intersection-over-Union (IoU) matching strategy between predicted bounding boxes $P_{\text{bbox}}$ and ground truth bounding boxes $GT_{\text{bbox}}$. Predicted boxes are considered True Positives ($TP_{\text{box}}$) if they match a ground truth box with an IoU score exceeding a predefined threshold $\tau$ (typically 0.5). False Positives ($FP_{\text{box}}$) are predicted boxes that do not match any ground truth box. The precision score is calculated as the ratio of True Positives to the total number of predicted boxes.

$$R_{\text{IoU Precision}}(y, GT) = \frac{TP_{\text{box}}}{TP_{\text{box}} + FP_{\text{box}}} = \frac{|TP_{\text{box}}|}{|P_{\text{bbox}}|}. \tag{7}$$

**IoU Recall Reward** This reward evaluates the model's ability to detect all text instances in the image. False Negatives ($FN_{\text{box}}$) are ground truth boxes that are not matched by any predicted box. The recall score is calculated as the ratio of True Positives to the total number of ground truth boxes.

$$R_{\text{IoU Recall}}(y, GT) = \frac{TP_{\text{box}}}{TP_{\text{box}} + FN_{\text{box}}} = \frac{|TP_{\text{box}}|}{|GT_{\text{bbox}}|}. \tag{8}$$

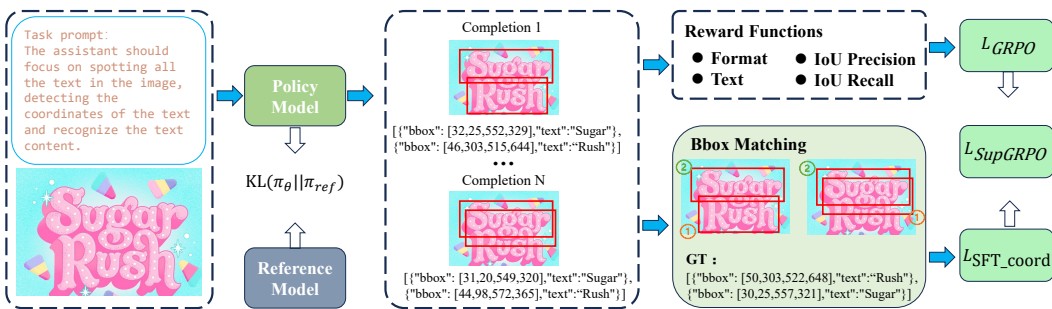

Figure 2: The overall framework of SupGRPO.

## 3.3 GRPO WITH MATCHING-BASED ONLINE SFT

While GRPO effectively optimizes the policy $\pi_\theta$ based on global reward signals, providing coarse-grained guidance for text spotting, it inherently lacks the fine-grained, token-level supervision necessary for precise coordinate prediction. Standard SFT, conversely, can provide this detailed supervision. However, applying standard SFT to the full output token sequence of text spotting results is problematic, as it imposes an arbitrary linear order on typically independent text instances and introduces irrelevant learning objectives related to sequence ordering rather than accurate spatial and content prediction for each instance.

To overcome the limitations of both paradigms, we propose **SupGRPO**, a novel approach that integrates matching-based online SFT specifically targeting the coordinate tokens within the GRPO framework. SupGRPO operates during the GRPO training process, performing an auxiliary supervised update based on sampled sequences. The overall framework of SupGRPO is shown in Fig. 2.

For each output sequence $y$ sampled from the current policy $\pi_\theta(\cdot|x)$ during the GRPO exploration step, we parse the predicted text instances, extracting their predicted bounding box coordinates $\{pb_1, \ldots, pb_m\}$ and associated text. We then establish correspondences by performing a matching process between these predicted bounding boxes $\{pb_1, \ldots, pb_m\}$ and the ground truth bounding boxes $\{gtb_1, \ldots, gtb_n\}$ for the input image $x$. The matching process is based on both text content and IoU. A match is considered successful only if the text content is identical and the IoU between the predicted and GT boxes is greater than 0. This results in a set of matched pairs $M = \{(pb_i, gtb_j)\}$.

For each predicted bounding box $pb_i$ that is successfully matched to a ground truth box $gtb_j$, we compute a supervised loss of Cross-Entropy on the predicted coordinate tokens. The total coordinate SFT loss for a sampled sequence $y$ and corresponding ground truth $GT(x)$ is the sum of negative log-likelihoods for the ground truth coordinate tokens of the matched instances, conditioned on the preceding generated tokens:

$$L_{\text{SFT-coord}}(y, GT(x)) = -\sum_M \sum_{t \in GT_{j,\text{coord}}} \log \pi_\theta(t|x, y_{<t}(t)), \qquad (9)$$

where $GT_{j,\text{coord}}$ denotes the sequence of ground truth coordinate tokens for the $j$-th instance, and $y_{<t}(t)$ represents the context from the sampled sequence $y$ preceding token $t$.

The overall training objective in SupGRPO combines the GRPO objective $\mathcal{J}_{\text{GRPO}}(\theta)$ (Equ. 1), which aims to maximize expected rewards, with minimizing the expected supervised coordinate loss $L_{\text{SFT-coord}}$. This is achieved by optimizing a combined objective function. Formulating this as minimizing a loss $\mathcal{L}_{\text{SupGRPO}}(\theta)$, the objective is:

$$\mathcal{L}_{\text{SupGRPO}}(\theta) = -\mathcal{J}_{\text{GRPO}}(\theta) + \lambda \cdot \mathbb{E}_{x \sim \mathcal{D}_{\text{GRPO}}, y \sim \pi_\theta(\cdot|x)}[L_{\text{SFT-coord}}(y, GT(x))], \qquad (10)$$

where $\lambda$ is a weighting hyperparameter balancing the two objectives and we set it as 1e-4 by default. This formulation ensures that the model is trained via policy gradients guided by task-specific rewards while simultaneously receiving direct, token-level supervision for spatial accuracy based on matching, effectively integrating the benefits of SFT for localization within the GRPO policy learning framework and mitigating the issues associated with standard sequence-based SFT.

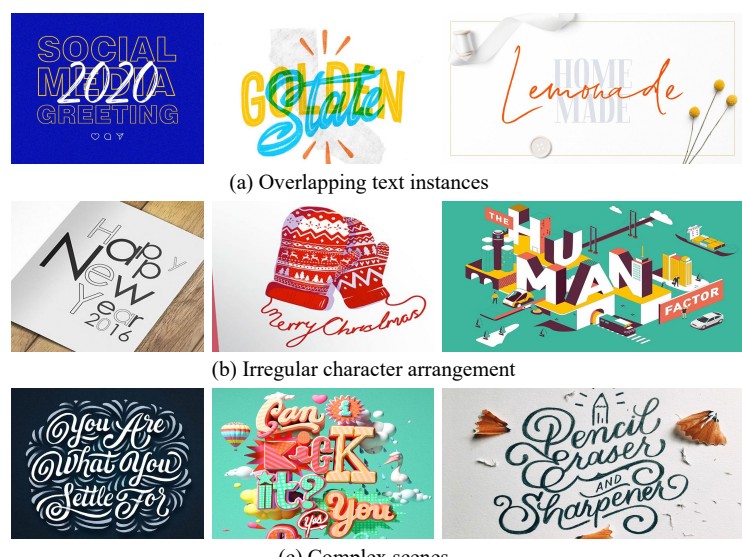

(a) Overlapping text instances

(b) Irregular character arrangement

(c) Complex scenes

Figure 3: Examples of different types for artistic text spotting from our ATS dataset

## 4 EXPERIMENTS

### 4.1 ARTISTIC TEXT SPOTTING DATASET

To benchmark the performance of different models on the artistic text spotting task, we construct a dataset of artistic text named ATS. The samples in this dataset are derived from two existing text segmentation datasets TextSeg (Xu et al., 2021) and WAS (Xie et al., 2024), from which we manually curated the artistic text samples, as well as corrected and re-annotated erroneous OCR labels. It includes 6500 training images and 2500 testing images. We provide word-level quadrilateral bounding box annotations and corresponding text transcriptions. The qualitative presentation of the ATS dataset is shown in Fig. 3. The data statistics are presented in the appendix, Fig. 4. Specifically, there are some special challenges compared to scene text spotting: **(1)** Many greeting cards and advertisements often have severe overlap between text instances, which poses great difficulties for text detection and recognition. **(2)** The irregular arrangement of characters makes it difficult for the model to judge which characters belong to the same word to get accurate detection results, and variations in the reading order also affect the recognition performance. **(3)** An artistic text image often perfectly combines multiple text instances and design patterns, which hardly exist in scene text images. This phenomenon results in complex relationships between instances and is difficult to distinguish from other elements. These challenges in complex scenarios require models with strong visual understanding and reasoning capabilities to detect and recognize text.

### 4.2 IMPLEMENTATION DETAILS

To verify the scalability and generalizability, we apply our SupGRPO on different architectures and scales. Specifically, we utilize the 3B and 7B variants of Qwen2.5-VL (Bai et al., 2025b), along with Qwen3-VL-8B (Bai et al., 2025a), as base models for LoRA fine-tuning. We designate the fine-tuned models as **TS-VL** (**T**ext **S**potting **V**ision-**L**anguage model). Our training is performed based on the open-source framework Open-R1 (Face, 2025) and its multimodal application, VLM-R1 (Shen et al., 2025). We conduct LoRA fine-tuning for one epoch utilizing 4 NVIDIA A100 GPUs, with a batch size of two samples per device. We configure the number of sampled output sequences for GRPO to 8 and set the maximum output length to 1024 tokens. The initial learning rate is set to 1e-6, and the ratio $\beta$ for the KL divergence regularization term is set to 0.04. For the training data, we combine the training sets of five datasets: our proposed ATS, Total-Text (Ch'ng et al., 2020), ICDAR 2015 (Karatzas et al., 2015), CTW1500 (Liu et al., 2019), and ReCTS (Zhang et al., 2019). To facilitate training, we reorganized the format of this data and added a task description to each entry, resulting in a total of approximately 30,000 training samples.

Table 1: Recognition results on scene text datasets of Total-Text, ICDAR 2015, SCUT-CTW1500 and artistic text dataset ATS. The best results are shown in bold font.

| Method | Total-Text | | ICDAR 2015 | | | CTW1500 | | ATS |
| | None | Full | S | W | G | None | Full | None |
|---|---|---|---|---|---|---|---|---|
| Mask TextSpotter v3 (Liao et al., 2020) | 71.2 | 78.4 | 83.3 | 78.1 | 74.2 | — | — | 56.9 |
| ABCNet v2 (Liu et al., 2021) | 70.4 | 78.1 | 82.7 | 78.5 | 73.0 | 57.5 | 77.2 | 53.4 |
| GLASS (Ronen et al., 2022) | 79.9 | 86.2 | 84.7 | 80.1 | 76.3 | — | — | — |
| TESTR (Zhang et al., 2022) | 73.3 | 83.9 | 85.2 | 79.4 | 73.6 | 56.0 | 81.5 | 55.0 |
| SwinTextSpotter (Huang et al., 2022) | 74.3 | 84.1 | 77.3 | 70.5 | — | 51.8 | 77.0 | 55.4 |
| SPTS (Peng et al., 2022) | 74.2 | 82.4 | 77.5 | 70.2 | 65.8 | 63.6 | 83.8 | 65.3 |
| TTS (Kittenplon et al., 2022) | 78.2 | 86.3 | 85.2 | 81.7 | 77.4 | — | — | — |
| DeepSolo (Ye et al., 2023) | 79.7 | 87.0 | 86.8 | 81.9 | 76.9 | 64.2 | 81.4 | 64.8 |
| Bridge (Huang et al., 2024) | 83.3 | 88.3 | 89.1 | 84.2 | 80.4 | 69.8 | 83.9 | 67.2 |
| Qwen2.5-VL-7B (Bai et al., 2025b) | 78.1 | 87.0 | 89.0 | 83.5 | 79.2 | 73.2 | 80.4 | 72.7 |
| InternVL3.5-8B (Wang et al., 2025) | 81.7 | 86.7 | 87.8 | 84.7 | 81.5 | 71.5 | 78.5 | 72.1 |
| Gemini 1.5-Pro (Reid et al., 2024) | 78.3 | 87.5 | 88.1 | 84.0 | 80.2 | 70.8 | 79.3 | 70.0 |
| TextMonkey (Liu et al., 2024) | 67.4 | 76.9 | 72.9 | 66.7 | 60.6 | 55.2 | 76.5 | 50.9 |
| GOT-OCR (Wei et al., 2024b) | 70.3 | 79.2 | 78.9 | 73.2 | 69.8 | 59.5 | 80.2 | 56.7 |
| Ocean-OCR (Chen et al., 2025) | 60.9 | 70.4 | 68.2 | 61.7 | 56.9 | 42.5 | 68.4 | 40.2 |
| Qwen2.5-VL-7B (SFT) | 82.6 | 86.4 | 86.8 | 83.6 | 81.4 | 79.2 | 85.6 | 86.2 |
| Qwen2.5-VL-7B (GRPO) | 84.2 | 88.3 | 86.9 | 85.5 | 84.4 | 82.4 | 87.7 | 87.9 |
| Two-Stage (SFT → GRPO) | 84.5 | 86.2 | 87.1 | 85.9 | 82.5 | 83.0 | 85.7 | 87.2 |
| TS-VL-3B (Qwen2.5-VL-3B) | 92.4 | 90.3 | 93.2 | 87.6 | 84.7 | 81.8 | 88.2 | 84.4 |
| TS-VL-7B (Qwen2.5-VL-7B) | 88.2 | 92.3 | 94.4 | 89.9 | 86.7 | 86.2 | 91.7 | 89.6 |
| TS-VL-8B (Qwen3-VL-8B) | **91.0** | **93.4** | **95.1** | **90.2** | **89.1** | **89.9** | **92.5** | **92.4** |

## 4.3 RESULTS OF TEXT SPOTTING

We evaluate the performance of our proposed method on four text spotting benchmarks: Total-Text (Ch'ng et al., 2020), ICDAR 2015 (Karatzas et al., 2015), CTW1500 (Liu et al., 2019), and our newly constructed ATS dataset. We assess both end-to-end text recognition and text detection performance. Comparisons are made against two main categories of models: specialized text spotting methods and other MLLMs. All the specialized models we compared against are fine-tuned on the ATS dataset. Besides, we independently fine-tuned the vanilla Qwen2.5-VL-7B using SFT and GRPO, employing the identical training dataset used for TS-VL. Experimental results demonstrate the effectiveness of our SupGRPO method (Tab. 1, Tab. 2 and Tab. 3).

For specialized text spotters, recognition is strictly coupled with detection (i.e., a detection failure inherently results in a recognition failure). In contrast, for MLLM-based methods, text recognition does not inherently rely on the successful localization by a dedicated detection module within the same framework. MLLMs can leverage global context to recognize text content even if the specific localization (bounding box) is less precise. Therefore, for these MLLMs, we directly evaluate the text recognition performance using the F1 score based on the recognized text strings from the full image. Following the common practice in previously published literature (Huang et al., 2024; Ye et al., 2023; Peng et al., 2022), which often compares methods with different evaluation pipelines (SPTS (Peng et al., 2022), TTS (Kittenplon et al., 2022), CRAFTS (Baek et al., 2020), Bridge (Huang et al., 2024)) within the same table, we report the overall recognition results in Tab. 1. For text detection evaluation, we strictly follow existing standard evaluation protocols. The results for text detection are presented in Tab. 2 and Tab. 3, respectively. Note that GOT-OCR (Wei et al., 2024b) and Ocean-OCR (Chen et al., 2025) lack inherent text detection capabilities, and TextMonkey (Liu et al., 2024) possesses only preliminary detection abilities.

The results of our TS-VL with SupGRPO significantly outperform those achieved by using SFT or GRPO alone. Notably, for complex scenes such as those in our ATS dataset, the performance improvement is particularly significant. Using a two-stage training approach (SFT followed by GRPO) is also inferior to our end-to-end joint training. It causes the model to optimize specific

capabilities in isolation rather than achieving mutual promotion, and it fails to resolve the instance order dependency faced by SFT and the reward sparsity issue associated with GRPO. The two-stage process inevitably increases training pipeline complexity and time overhead.

Regarding text detection, our method shows a substantial improvement compared to the baseline generic MLLMs (as evidenced in Tab. 2 and Tab. 3). However, most MLLMs still exhibit a performance gap compared to specialized models explicitly optimized for detection accuracy. To further enhance detection performance, we fine-tuned an advanced MLLM, Qwen3-VL-8B, and the results are comparable to those of specialized models.

Table 2: Detection performance on ATS.

| Method | Detection |
|---|---|
| ABCNet v2 | 87.5 |
| ABINet++ | **88.2** |
| TESTR | 86.4 |
| SwinTextSpotter | 85.9 |
| DeepSolo | 86.7 |
| Qwen2.5-VL-7B | 27.4 |
| InternVL3.5-8B | 21.2 |
| Gemini 1.5-Pro | 26.8 |
| TextMonkey | 17.5 |
| Qwen2.5-VL-7B (SFT) | 72.5 |
| Qwen2.5-VL-7B (GRPO) | 69.6 |
| Two-Stage (SFT → GRPO) | 70.6 |
| TS-VL-3B | 71.6 |
| TS-VL-7B | 77.1 |
| TS-VL-8B | 86.2 |

Table 3: Detection performance on scene text datasets of Total-Text and ICDAR 2015.

| Method | Total | IC15 |
|---|---|---|
| TextPerceptron | 85.2 | 87.1 |
| PGNet | 86.1 | 88.2 |
| ABCNet v2 | 87.0 | 88.1 |
| DeepSolo | **87.3** | **90.0** |
| Qwen2.5-VL-7B | 23.2 | 19.2 |
| InternVL3.5-8B | 17.8 | 16.5 |
| Gemini 1.5-Pro | 21.7 | 18.7 |
| TextMonkey | 11.7 | 10.2 |
| Qwen2.5-VL-7B (SFT) | 65.2 | 68.6 |
| Qwen2.5-VL-7B (GRPO) | 60.5 | 63.6 |
| Two-Stage (SFT → GRPO) | 66.1 | 67.3 |
| TS-VL-3B | 68.4 | 71.6 |
| TS-VL-7B | 70.1 | 73.8 |
| TS-VL-8B | 84.0 | 86.4 |

## 4.4 ABLATION STUDY

In this section, we conduct ablation studies on the artistic text spotting dataset ATS to validate the effectiveness of our proposed designs. For faster validation of the effectiveness of our various designs, we chose Qwen2.5-VL-3B (Bai et al., 2025b) as the baseline for the ablation experiments.

**ATS dataset.** To specifically quantify the impact of incorporating the ATS dataset into the training mix, Tab. 4 compares the performance on various text spotting benchmarks with and without the inclusion of ATS training data. This validates that the ATS dataset provides high-quality, diverse training samples that enhance the model's overall generalization capability.

**Training strategy.** We train the baseline independently using SFT and GRPO with the same dataset. The results in Tab. 5 reveal that SFT yields more significant improvements in text detection performance compared to GRPO, whereas GRPO provides a more substantial boost to text recognition performance than SFT. This observation directly motivated our approach of jointly training with both SFT and GRPO. Besides, we applied data augmentation for SFT. While it yields minor gains, it still significantly underperforms SupGRPO.

**Tokens for SFT.** Furthermore, when jointly applying online SFT with GRPO, optimizing different types of tokens for SFT significantly affects the detection and recognition results, as shown in Tab. 6. Applying SFT to the entire sequence of all tokens yields some improvement in overall performance. However, the fixed and unique order in which text instances are arranged within the GT creates an order prior that interferes with word-level detection and recognition. Alternatively, applying a mask to the token sequence to supervise only the text content tokens can improve text recognition performance but weaken text detection performance. Conversely, supervising only the location tokens and performing matching on them avoids the interference of word order, leading to comprehensive improvements in both detection and recognition performance.

**Matching mechanism.** For text box matching, we explore three distinct approaches. The first is an IoU-based method: for each predicted box, it is matched to the GT box with the largest IoU,

Table 4: Ablation study on the effectiveness of introducing the ATS dataset during training.

| Training Data | Total (Det / Rec) | IC15 (Det / Rec) | ATS (Det / Rec) | CTW (Rec) |
|---|---|---|---|---|
| Without ATS | 65.8 / 83.9 | 67.2 / 82.8 | 65.3 / 75.7 | 77.9 |
| With ATS (Ours) | 68.4 / 86.0 | 71.6 / 84.7 | 71.6 / 84.4 | 81.8 |

provided that IoU is greater than 0.3. If a predicted box's IoU with all GT boxes is less than 0.3, its loss is not calculated. This method can easily miss many valid predicted boxes or incorrectly match them to boxes of other text instances. The second is a text-based method, which matches predicted boxes to their corresponding GT boxes solely based on identical text content. Leveraging MLLM's powerful text recognition capability, this text-based approach can achieve a high match rate for predicted boxes. However, this method struggles to distinguish between multiple text instances with the same content within a single image, potentially leading to incorrect GT box matches. Our proposed method, in contrast, performs matching based on both text content and IoU. A match is considered successful only if the text content is identical and the IoU between the predicted and GT boxes is greater than 0. Tab. 7 demonstrates the effectiveness of this approach.

**Reward functions.** Finally, we conduct ablation studies on the design of the rule-based reward function for GRPO. Tab. 8 shows the results. Rewarding solely based on text recognition content (Row 2) leads to a decrease in text detection performance against the combined reward settings (Rows 3 and 4). While designing a single reward for detection that uses the harmonic mean (F1-score) to combine precision and recall substantially increases the complexity of the reward function. As a result, it led to poorer model convergence and lower sample efficiency. Therefore, treating precision and recall as two independent rewards allows us to improve overall performance further.

Table 5: Ablation study on training strategy.

| | Detection | Recognition |
|---|---|---|
| SFT | 63.3 | 77.1 |
| SFT + Aug | 65.1 | 77.4 |
| GRPO | 62.0 | 80.0 |
| SupGRPO | 71.6 | 84.4 |

Table 6: SupGRPO for different tokens.

| | Detection | Recognition |
|---|---|---|
| GRPO | 62.0 | 80.0 |
| + All Tokens SFT | 64.0 | 81.2 |
| + Text Tokens SFT | 60.8 | 84.1 |
| + Location Tokens SFT | 71.6 | 84.4 |

Table 7: Different matching mechanism for online SFT location tokens.

| | Detection | Recognition |
|---|---|---|
| All Tokens SFT | 64.0 | 81.2 |
| IoU-based | 66.5 | 81.7 |
| Text-based | 68.6 | 82.8 |
| IoU & Text | 71.6 | 84.4 |

Table 8: Ablation study on reward functions.

| | Detection | Recognition |
|---|---|---|
| Baseline | 26.3 | 70.2 |
| Text | 61.1 | 83.2 |
| Text & F | 69.3 | 83.5 |
| Text & P & R | 71.6 | 84.4 |

## 5 CONCLUSION

This paper addressed the challenge of text spotting in complex artistic images, where existing MLLMs often lack precise localization. We explored applying the GRPO training paradigm to text spotting for the first time, designing specific rewards. To overcome GRPO's lack of fine-grained coordinate supervision and standard SFT's sequence issues, we proposed SupGRPO, which combines GRPO with a matching-based online SFT specifically targeting coordinate tokens. We also contributed the ATS dataset for evaluating performance on artistic text. Our experiments demonstrated SupGRPO's superior performance in both text detection and end-to-end recognition, effectively integrating policy optimization and targeted coordinate learning to advance text spotting. Finally, we believe that our joint training strategy may offer valuable new insights and a potential training paradigm for the broader MLLM research community.

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

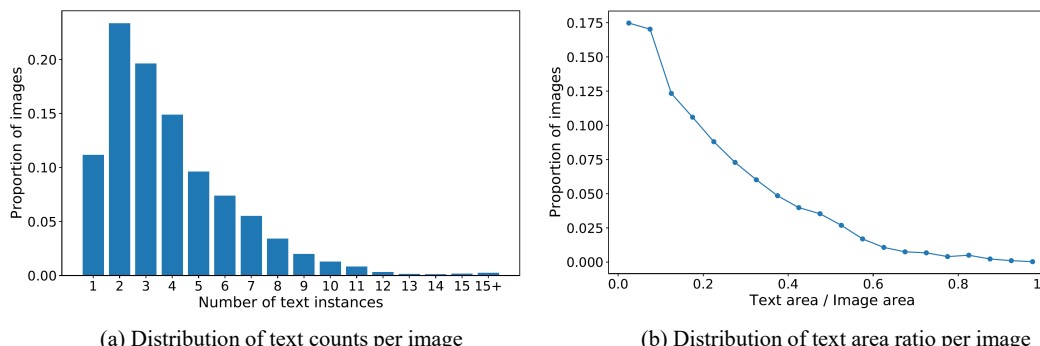

(a) Distribution of text counts per image  (b) Distribution of text area ratio per image

Figure 4: Data statistics of the proposed artistic text spotting dataset ATS.

## A  APPENDIX: DISTRIBUTION STATISTICS OF THE ATS DATASET

To further demonstrate more features of our proposed ATS dataset, we counted the number of text instances contained in each image, as shown in Fig. 4 (a). Unlike scene text images, which contain a large number of text instances, most artistic text images contain no more than 10 instances. Although a small number of instances reduces the difficulty of detection to a certain extent, it is still difficult for existing methods to achieve satisfactory recognition performance.

In addition, we statistically analyze the ratio of text regions to the overall area of each image, as shown in Fig. 4 (b). A significant portion of these images reveals that text content constitutes no more than 20% of the total image space. The remaining areas are predominantly comprised of rich artistic elements or other objects, thereby posing considerable challenges for artistic text spotting. Nevertheless, we can harness the capabilities of MLLMs to exploit the contextual information within these non-textual regions to facilitate the recognition of textual content.

## B  APPENDIX: TRAINING CURVE COMPARISON

This appendix provides a comparative analysis of the vanilla GRPO and our proposed SupGRPO methods during the training process. As can be clearly observed in Fig. 5, SupGRPO demonstrates superior stability and efficiency throughout training. Specifically, for each of the individual reward functions (Precision, Recall, Content, and Format Spotting Reward) as well as the Total Reward, the curve for SupGRPO (red line) is consistently and stably higher than that of the vanilla GRPO (blue line), indicating that it learns more effectively and achieves higher reward values. Furthermore, in the Total Loss graph, the loss curve for SupGRPO not only converges to a lower value but also exhibits significantly less fluctuation compared to the vanilla GRPO, which serves as evidence of a more stable training process for our method. Collectively, these curves show that by incorporating matching-based online SFT, SupGRPO effectively mitigates the issues present when using GRPO alone, leading to more stable and efficient model optimization.

## C  APPENDIX: QUALITATIVE COMPARISON

Fig. 6 illustrates a qualitative comparison of our model with other specialized text spotting models and MLLMs across four datasets: our proposed ATS, Total-Text (Ch'ng et al., 2020), ICDAR 2015 (Karatzas et al., 2015), and CTW1500 (Liu et al., 2019). Compared to existing MLLMs, our model demonstrates significantly superior text detection performance, yielding more accurate bounding boxes. Against specialized text spotting models, ours exhibits a better ability to perceive and understand artistic text, thereby accurately capturing all text content. Furthermore, leveraging the advantages of MLLMs, our model is capable of multi-language text spotting. Interestingly, our model can automatically infer and complete occluded text content, a phenomenon frequently observed in the ICDAR 2015 dataset.

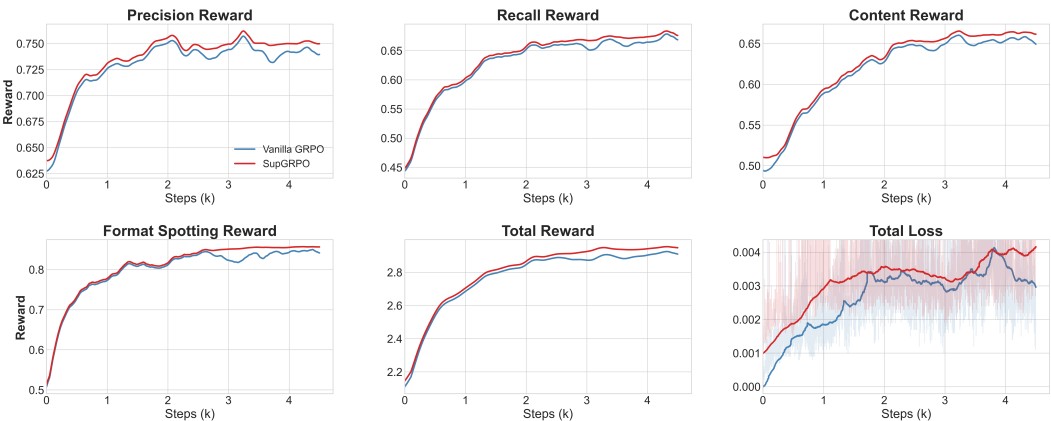

Figure 5: Training curve comparison between the vanilla GRPO and our SupGRPO.

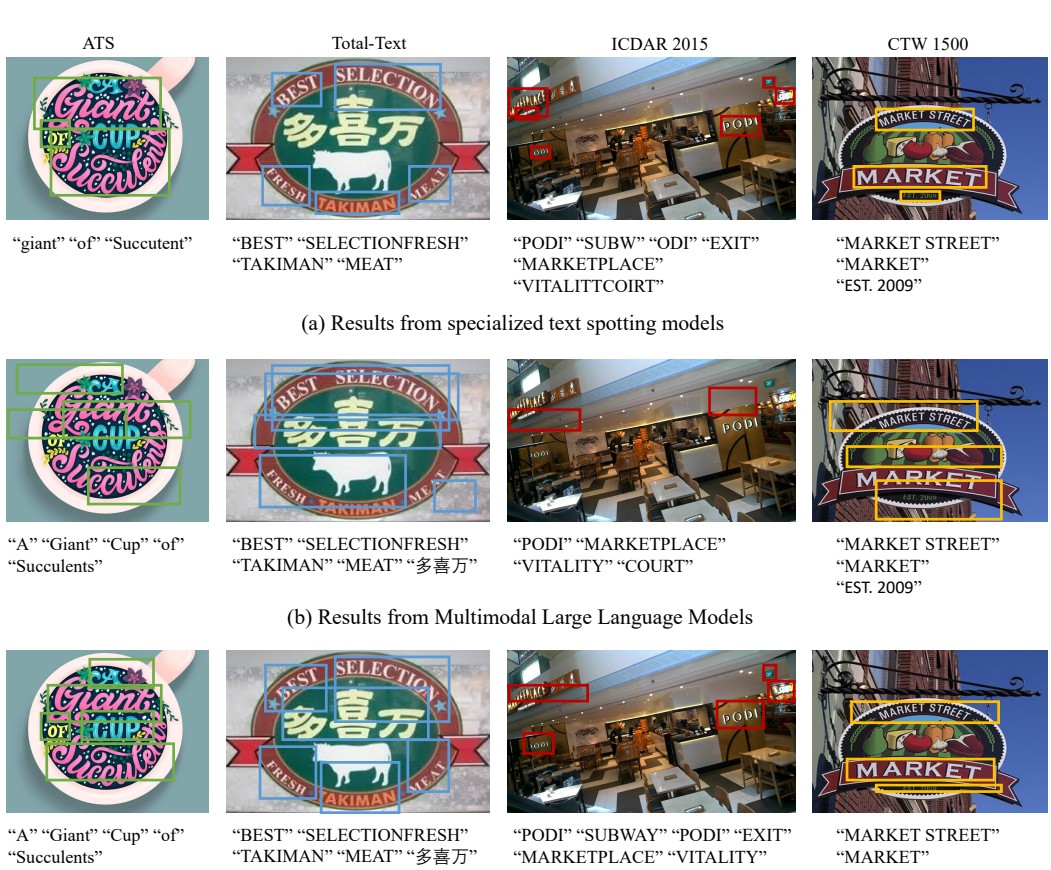

Figure 6: Qualitative comparison with other specialized text spotting models and MLLMs.

## D  LIMITATION

Despite the significant advancements achieved by SupGRPO, two limitations remain. First, while our optimized models (e.g., utilizing Qwen3-VL-8B) achieve detection performance comparable to specialized text spotting models, they do not significantly surpass them in pure localization precision. Second, fine-tuning on domain-specific text spotting data inevitably leads to a slight degradation in performance on general multimodal benchmarks, a common phenomenon known as catastrophic forgetting. However, this issue is effectively mitigated by our adoption of LoRA, which freezes the pre-trained backbone parameters to preserve general world knowledge while adapting to the target task.

