# OpenReview forum: "SupGRPO: Enhancing GRPO with Matching-based Online SFT for Text Spotting"
_ICLR.cc/2026/Conference — Submitted to ICLR 2026_

### Official Review · Reviewer_SV9w · 2025-10-30

**Soundness:** 3
**Presentation:** 3
**Contribution:** 3
**Rating:** 6
**Confidence:** 4

**Summary:**

This paper proposes SupGRPO for text spotting. The method performs joint training by jointly optimizing GRPO and matching-based online SFT, and designs the ATS dataset for artistic and complex scenes. Experiments show that the proposed method achieves excellent performance on multiple text spotting datasets.

**Strengths:**

1. This paper introduces GRPO into the text spotting task for the first time and designs four task-specific reward functions: format, text content, IoU precision, and IoU recall.
2. The authors constructed an artistic text dataset containing approximately 9K samples to expand the applicability of existing datasets in artistic and complex scenes.
3. Experimental results show that introducing SupGRPO significantly improves the model’s performance on the text spotting task.

**Weaknesses:**

1. SupGRPO is trained on a dataset that includes ATS, which mainly contains artistic text and differs from other datasets that focus on natural scenes. The compared models are not fine-tuned on this dataset. We suggest fine-tuning the baselines on the same dataset to verify the method’s effectiveness.
2. Although the authors selected strong general-purpose MLLMs (e.g., Qwen2.5-VL, InternVL 3.5) as baselines, the experiments still lack comparisons with OCR-optimized multimodal models (e.g., Ocean-OCR, TextMonkey). It is recommended to include such comparisons to more comprehensively verify the performance and competitiveness of SupGRPO in text spotting.
3. In the ablation study, the authors claim that “rewarding only text content degrades detection performance,” yet Table 7 shows that the detection score increases significantly from 26.3 to 61.1 (Baseline → Text). This inconsistency should be clarified by explaining the evaluation setup or experimental configuration.

**Questions:**

Please refer to weaknesses 1 and 2.

---

> ### Author Response · Authors · 2025-11-24
>
> 1. We emphasize that the experimental comparisons in our paper are **strictly fair and consistent**. First, we fine-tuned the baseline MLLM using three different training strategies: SFT, GRPO, and SupGRPO. As shown in Tables 1, 2, and 3, all three experiments utilized the identical training data, including the ATS dataset, validating that SupGRPO is optimal. In addition to the fine-tuned baseline MLLM, all the specialized models we compared against were also fine-tuned on the ATS dataset. Therefore, the experiments presented in the paper represent a fully fair comparison. Additionally, we have provided an ablation study to compare the model performance with and without the inclusion of ATS training data (details shown in Response 3 to Reviewer KoR9).
>
> 2. We have incorporated comparisons with three OCR-optimized multimodal models: GOT-OCR 2.0, Ocean-OCR, and TextMonkey, as presented in the tables below. It is important to note that GOT-OCR 2.0 and Ocean-OCR lack inherent text detection capabilities, and TextMonkey possesses only preliminary detection abilities. Consequently, their performance significantly lags behind our model.
>
>     | Recognition | Total (None) | Total (Full) | IC15 (S) | IC15 (W) | IC15 (G) | CTW (None) | CTW (Full) | ATS (None) |
>     | :--- | :---: | :---: | :---: | :---: | :---: | :---: | :---: | :---: |
>     | Ours | **88.2** | **92.3** | **94.4** | **89.9** | **86.7** | **86.2** | **91.7** | **89.6** |
>     | TextMonkey | 67.4 | 76.9 | 72.9 | 66.7 | 60.6 | 55.2 | 76.5 | 50.9 |
>     | GOT-OCR2.0 | 70.3 | 79.2 | 78.9 | 73.2 | 69.8 | 59.5 | 80.2 | 56.7 |
>     | Ocean-OCR | 60.9 | 70.4 | 68.2 | 61.7 | 56.9 | 42.5 | 68.4 | 40.2 |
>
>
>     | Detection | ATS | Total | IC15 |
>     | :--- | :--- | :--- | :--- |
>     | Ours | 77.1 | 70.1 | 73.8 |
>     | TextMonkey | 17.5 | 11.7 | 10.2 |
>
>
> 3. We apologize for the misunderstanding caused by the phrasing. We clarify that the "Baseline" in Table 7 refers to the vanilla Qwen2.5-VL-3B model without any reinforcement learning training. The statement that "rewarding only text content degrades detection performance" was intended to compare the **Text-only reward setting** (Row 2) against the **combined reward settings** (Rows 3 and 4). Applying a text reward significantly underperforms compared to the setting that includes explicit spatial rewards (61.1 vs. 71.6). Thus, relying *solely* on text rewards leads to suboptimal detection performance relative to our full SupGRPO method.

---

> > ### Comment · Reviewer_SV9w · 2025-11-27
> >
> > I appreciate the authors' rebuttal, which addresses all my concerns, and I will maintain my positive rating.

---

### Official Review · Reviewer_p8Ee · 2025-10-30

**Soundness:** 3
**Presentation:** 3
**Contribution:** 3
**Rating:** 4
**Confidence:** 4

**Summary:**

This paper proposes SupGRPO, a method for text spotting that combines both SFT and GRPO. In SupGRPO, the authors introduce a matching mechanism before SFT to reduce the instance order dependency problem. They also design several text-spotting–specific reward functions for GRPO. Furthermore, the authors construct an ATS dataset for training and evaluation. Experimental results demonstrate that their method outperforms other compared MLLMs.

**Strengths:**

1.This paper analyzes the drawbacks of the SFT and GRPO frameworks in the text spotting task. To address these issues, the authors propose a combined training method that achieves better performance.
2.Experimental results show that the proposed method outperforms other general MLLMs on both text recognition and detection tasks.
3.The authors create a new and challenging text spotting dataset, ATS, for evaluation.
4.The paper is well organized and clearly presented.

**Weaknesses:**

1.The authors show that their method achieves significant improvement on the new ATS dataset. However, their model has been trained on the ATS training set, which is unfair. The authors should provide results without training on ATS or fine-tune other models on the ATS for a fair comparison.
2.Only general MLLMs are compared. The authors should further include comparisons with MLLMs specifically designed for OCR tasks, such as GOT-OCR 2.0.
3. Why should SFT and GRPO be trained jointly in a single stage? A comparison with the commonly used two-stage training approach (SFT followed by GRPO) should be included as a baseline.
4.To better verify the effectiveness of their method, the authors should evaluate models with different scales and architectures.

**Questions:**

Answer the questions in Weakness.

---

> ### Author Response · Authors · 2025-11-24
>
> 1. We emphasize that the experimental comparisons in our paper are **strictly fair and consistent**. First, we fine-tuned the baseline MLLM using three different training strategies: SFT, GRPO, and SupGRPO. As shown in Tables 1, 2, and 3, all three experiments utilized the identical training data, including the ATS dataset, validating that SupGRPO is optimal. In addition to the fine-tuned baseline MLLM, all the specialized models we compared against were also fine-tuned on the ATS dataset. Therefore, the experiments presented in the paper represent a fully fair comparison. Additionally, we have provided an ablation study to compare the model performance with and without the inclusion of ATS training data (details shown in Response 3 to Reviewer KoR9).
>
> 2. We have incorporated comparisons with three OCR-optimized multimodal models: GOT-OCR 2.0, Ocean-OCR, and TextMonkey, as presented in the tables below. It is important to note that GOT-OCR 2.0 and Ocean-OCR lack inherent text detection capabilities, and TextMonkey possesses only preliminary detection abilities. Consequently, their performance significantly lags behind our model.
>
>     | Recognition | Total (None) | Total (Full) | IC15 (S) | IC15 (W) | IC15 (G) | CTW (None) | CTW (Full) | ATS (None) |
>     | :--- | :---: | :---: | :---: | :---: | :---: | :---: | :---: | :---: |
>     | Ours | **88.2** | **92.3** | **94.4** | **89.9** | **86.7** | **86.2** | **91.7** | **89.6** |
>     | TextMonkey | 67.4 | 76.9 | 72.9 | 66.7 | 60.6 | 55.2 | 76.5 | 50.9 |
>     | GOT-OCR2.0 | 70.3 | 79.2 | 78.9 | 73.2 | 69.8 | 59.5 | 80.2 | 56.7 |
>     | Ocean-OCR | 60.9 | 70.4 | 68.2 | 61.7 | 56.9 | 42.5 | 68.4 | 40.2 |
>
>
>
>     | Detection | ATS | Total | IC15 |
>     | :--- | :--- | :--- | :--- |
>     | Ours | 77.1 | 70.1 | 73.8 |
>     | TextMonkey | 17.5 | 11.7 | 10.2 |
>
>
>
> 3. We added an experiment using a two-stage training approach (SFT followed by GRPO). As shown in the table below, its performance is inferior to our end-to-end joint training. We attribute this to three key reasons:
>
>     * Two-stage training causes the model to optimize specific capabilities in isolation rather than achieving mutual promotion. As observed in our ablation study, SFT primarily boosts detection while GRPO favors recognition. Separating these stages prevents the two tasks from reinforcing each other. Joint training allows the mutual promotion.
>     * The two-stage approach cannot address the two fundamental issues identified in our paper. The initial SFT stage introduces **Instance Order Dependency** and the second GRPO stage faces **Reward Sparsity**. Our SupGRPO can perfectly solve both issues at the same time.
>     * The two-stage process inevitably increases training pipeline complexity and time overhead. Our single-stage joint training offers a more efficient and unified solution.
>
>
>     | Recognition | Total (None) | Total (Full) | IC15 (S) | IC15 (W) | IC15 (G) | CTW (None) | CTW (Full) | ATS (None) |
>     | :--- | :---: | :---: | :---: | :---: | :---: | :---: | :---: | :---: |
>     | Joint Training (SupGRPO) | 88.2 | 92.3 | 94.4 | 89.9 | 86.7 | 86.2 | 91.7 | 89.6 |
>     | Two-Stage (SFT $\to$ GRPO) | 84.5 | 86.2 | 87.1 | 85.9 | 82.5 | 83.0 | 85.7 | 87.2 |
>
>
>
>     | Detection | ATS | Total | IC15 |
>     | :--- | :--- | :--- | :--- |
>     | Joint Training (SupGRPO) | 77.1 | 70.1 | 73.8 |
>     | Two-Stage (SFT $\to$ GRPO) | 70.6 | 66.1 | 67.3 |
>
>
>
> 4. We appreciate the suggestion to verify scalability and generalizability. In our original paper, we evaluated Qwen2.5-VL at two different scales: the 7B model for main results and the 3B model for ablation studies, demonstrating consistent effectiveness across these sizes. To further validate our method on a different architecture and larger scale, we conducted an additional experiment using the state-of-the-art Qwen3-VL-8B. As shown in the table below, SupGRPO successfully generalizes to this new architecture, achieving significant performance gains. This confirms that our proposed strategy is robust and effective across varying model scales and architectures.
>
>     | Recognition | Total (None) | Total (Full) | IC15 (S) | IC15 (W) | IC15 (G) | CTW (None) | CTW (Full) | ATS (None) |
>     | :--- | :---: | :---: | :---: | :---: | :---: | :---: | :---: | :---: |
>     | Qwen2.5-VL-7B | 88.2 | 92.3 | 94.4 | 89.9 | 86.7 | 86.2 | 91.7 | 89.6 |
>     | Qwen2.5-VL-3B | 86.0 | 90.3 | 93.2 | 87.6 | 84.7 | 81.8 | 88.2 | 84.4 |
>     | Qwen3-VL-8B | 91.0 | 93.4 | 95.1 | 90.2 | 89.1 | 89.9 | 92.5 | 92.4 |
>
>
>     | Detection | ATS | Total | IC15 |
>     | :--- | :--- | :--- | :--- |
>     | Qwen2.5-VL-7B | 77.1 | 70.1 | 73.8 |
>     | Qwen2.5-VL-3B | 71.6 | 68.4 | 71.6 |
>     | Qwen3-VL-8B | 86.2 | 84.0 | 86.4 |

---

### Official Review · Reviewer_KoR9 · 2025-10-31

**Soundness:** 3
**Presentation:** 3
**Contribution:** 2
**Rating:** 4
**Confidence:** 5

**Summary:**

This work introduces SupGRPO, which GRPO with SFT to improve the text spotting capabilities of multimodal large language models. Additionally, the paper presents the ATS dataset comprising artistic text images. Experiments across multiple text spotting benchmarks validate the effectiveness of the proposed method.

**Strengths:**

- This paper is clearly written and easy to follow.
- This paper gives a successful try of GRPO on text spotting.
- This paper provides interesting observations in applying SFT and GRPO to text spotting.

**Weaknesses:**

- The contribution is somewhat limited. Authors apply SFT plus GRPO to text spotting and modify the rewards. They re-annotate erroneous OCR labels in TextSeg and WAS. However, the influence of the annotation error is not shown.
- Applying RL for a sole text spotting task is less meaningful.
- The effectiveness of introducing ATS during training for other benchmarks is not validated.
- The detection performance lags far behind existing text spotting specialists.

**Questions:**

- Considering the text reward, why use the word-level F1-score which falls short on reflecting the error degree of recognition. Is ANLS or edit distance better?
- What about comparing SupGRPO to standard supervised learning with extensive data augmentation? Would this reduce the performance gap attributed to GRPO?
- The paper lacks ablation studies to show the influence of introducing ATS training data for other benchmarks.

---

> ### Author Response · Authors · 2025-11-24
>
> 1. We respectfully disagree that the contribution is limited. We clarify the significant value of our work from three key aspects:
>
>     *  To the best of our knowledge, this work is the first to expand the capability boundaries of MLLMs specifically for the text spotting task. Furthermore, it is the first academic study focused on the challenging scenario of artistic text spotting, moving beyond standard scene text.
>     *  SupGRPO is not a trivial superposition of SFT and GRPO. It introduces an end-to-end fusion strategy with a specially designed matching-based online SFT mechanism. This design resolves the **instance order dependency** of standard SFT and the **reward sparsity** of vanilla GRPO. By doing so, we also achieve a synergy where SFT enhances detection precision while GRPO boosts recognition reasoning, results that cannot be achieved by either method alone.
>     *  We constructed a standardized dataset for artistic text spotting. We performed comprehensive evaluations of both specialized models and MLLMs on this benchmark, establishing a critical reference point for future research in this domain.
>
>     Regarding the influence of re-annotating erroneous labels: We emphasize that the original labels in TextSeg and WAS contained errors and were often missing word-level bounding boxes. Using such data would be detrimental in two ways: training on erroneous labels degrades model performance, and more critically, evaluating on incorrect ground truth leads to scientifically invalid metrics. Therefore, rather than showing the negative influence of noise, we focused on establishing a valid and clean benchmark for accurate evaluation.
>
> 2. We respectfully argue that applying RL is meaningful and critical for complex text spotting. First, standard SFT suffers from the instance order dependency problem, where the model is forced to learn an arbitrary sequence order for independent text instances. RL (GRPO) effectively resolves this by optimizing for the *set* of predictions via order-agnostic rewards. Second, RL enables the direct optimization of non-differentiable evaluation metrics (IoU and F1-score) and encourages reasoning exploration in ambiguous artistic scenes. Our ablation study supports this necessity, showing that GRPO significantly outperforms SFT in recognition by exploring diverse reasoning paths that SFT cannot capture. Finally, we believe that our joint training strategy may offer valuable new insights and a potential training paradigm for the broader MLLM research community.
>
> 3. Following the experimental setup described in our ablation study, we conducted an additional experiment to specifically quantify the impact of incorporating the ATS dataset into the training mix. The table below compares the performance on various text spotting benchmarks. This validates that the ATS dataset provides high-quality, diverse training samples that enhance the model's overall generalization capability.
>
>     | Training Data | Total-Text (Det/Rec) | ICDAR 2015 (Det/Rec) | ATS (Det/Rec) | CTW (Rec) |
>     | :--- | :--- | :--- | :--- | :--- |
>     | Without ATS | 65.8 / 83.9 | 67.2 / 82.8 | 65.3 / 75.7 | 77.9 |
>     | With ATS (Ours) | 68.4 / 86.0 | 71.6 / 84.7 | 71.6 / 84.4 | 81.8 |
>
> 4. The response to this issue is shown in Response 1 to Reviewer oBmt. The new experimental results achieved the best detection performance.
>
> 5. We deliberately chose the word-level F1-score to strictly align with the standard evaluation protocols for text spotting benchmarks (e.g., ICDAR 2015, Total Text), which typically rely on **Exact Match** criteria. Besides, a fundamental advantage of reinforcement learning is its ability to directly optimize the non-differentiable evaluation metrics used at test time. Using "soft" rewards like ANLS or Edit Distance introduces a misalignment: it encourages the model to learn "approximate" spellings (e.g., "Applo" vs. "Apple") to maximize partial rewards, whereas the test protocol penalizes any character error as a complete failure. Additionally, since our framework treats outputs as **unordered multisets** to resolve instance order dependency, calculating pair-wise metrics like ANLS would require computationally expensive matching steps, while the set-based F1-score is inherently efficient and order-agnostic.
>
> 6. Extensive data augmentation operates in the **input pixel space** and cannot address the fundamental structural limitations of SFT in the **output space**. Specifically, augmentation fails to resolve the **Instance Order Dependency** and cannot directly optimize non-differentiable metrics (IoU/F1) as SupGRPO does. We applied augmentation (rotation, color jittering, resize) to the baseline. While it yields minor gains, it still significantly underperforms SupGRPO.
>
>     | Method | Detection | Recognition |
>     | :--- | :--- | :--- |
>     | SFT (Baseline) | 63.3 | 77.1 |
>     | SFT + Augmentation | 65.1 | 77.4 |
>     | SupGRPO (Ours) | 71.6 | 84.4 |

---

### Official Review · Reviewer_oBmt · 2025-10-31

**Soundness:** 3
**Presentation:** 2
**Contribution:** 3
**Rating:** 6
**Confidence:** 4

**Summary:**

This paper aims to solve the problem of inaccurate text detection of MLLMs. First, they find that though SFT and GRPO can both improve the detection and recognition performance, SFT is superior to GRPO for detection but GRPO is superior to SFT for recognition. As a result, they combine GRPO and SFT to train MLLMs to better spotting text, and corresponding reward functions and a strategy that SFT only applies to matching coordinate tokens are designed. A dataset ATS is curated with existing datasets and extensive experiments are implemented.

**Strengths:**

1) Addressing the very low accuracy of text detection of MLLMs is valuable since in many scenarios both recognition and detection are important for real applications.
2) The proposed SupGRPO, which utilizes the specific Format Reward, Text Reward, IoU Precision Reward and IoU Recall Reward for policy optimization and simultaneously uses matching-based online SFT only for coordinate tokens, can boost both the detection and the recognition performance.
3) The detection performance of MLLMs can be improved with large margin by the proposed method, and the recognition performance can also be improved greatly. Especially, on ATS and CTW datasets, the recognition performance can be improved with large margin.

**Weaknesses:**

1) Though the detection accuracy is improved a lot, it is still can not be comparable with specialized text spotting models. Maybe the detection output of the specialized models is input to the MLLMs, the overall end-to-end recognition accuracy can be higher than that of MLLMs trained with SupGRPO, and the overall parameter amount and inference latency are comparable.
2) Tab. 1, Tab. 4-7, the recognition accuracy is not stated clearly. Is it the end-to-end recognition accuracy or the recognition accuracy of the cropped text region using the detection GT? If it is the end-to-end accuracy, the analysis for detection and recognition improvement is coupled together because end-to-end recognition accuracy includes detection’ performance implicitly. Please clarify it.
3) Eqa.6, do the predicted text strings and the ground truth text strings eliminate duplicate words? How to deal with duplicate words in the same image?
4) Fig.2 is not referred in the main body of the paper.
5) After trained with SupGRPO, does the performance of the MLLMs on other metrics except OCR degrade?

**Questions:**

See the weaknesses.

---

> ### Author Response · Authors · 2025-11-24
>
> 1. First, as noted in the **Limitation** section of our paper, we openly acknowledged that the detection performance of our initial implementation lagged behind specialized models. To address this gap and demonstrate the scalability of our approach, we conducted two supplementary experiments: (1)  **Full Fine-Tuning** on the original baseline (Qwen2.5-VL-7B), and (2)  **LoRA Fine-Tuning** on the state-of-the-art MLLM (Qwen3-VL-8B).
>
>     As shown in the table below, both experimental settings achieved detection accuracy that **competitive with** specialized models. This strongly validates the significant technical potential of MLLMs for high-performance text spotting.
>
>     | Method | ATS | Total | IC15 |
>     | :--- | :--- | :--- | :--- |
>     | DeepSolo (Specialized Model) | 86.7 | 87.3 | 90.0 |
>     | Qwen2.5-VL-7B + SupGRPO (LoRA) | 77.1 | 70.1 | 73.8 |
>     | **Qwen2.5-VL-7B + SupGRPO (Full)** | 88.0 | 89.6 | 89.8 |
>     | **Qwen3-VL-8B + SupGRPO (LoRA)** | 86.2 | 84.0 | 86.4 |
>
>     Furthermore, regarding your pragmatic suggestion of a pipeline approach (Specialized Detector + MLLM Recognizer), while we appreciate the engineering perspective, we believe our unified End-to-End SupGRPO offers fundamental advantages over a cascaded pipeline:
>
>     *  The pipeline is strictly bounded by the specialized detector's recall. If the detector fails on ambiguous artistic text (False Negatives), the MLLM never sees it. In contrast, ours leverages the MLLM's semantic reasoning to detect text that might be visually occluded but semantically probable, effectively using "reading" to improve "detecting."
>     *  The pipeline typically feeds cropped image regions to the recognizer, destroying the global visual context. SupGRPO processes the full image, utilizing global layout and scene information which is critical for deciphering complex artistic text.
>     *  Our "One Model" paradigm simplifies the deployment complexity of maintaining heterogeneous systems (Detector + LLM) and avoids the latency accumulation inherent in serial processing.
>
> 2. We clarify that the recognition accuracy reported in Table 1 and Tables 4-7 refers to End-to-End recognition accuracy (full-image text spotting), rather than recognition on cropped regions using ground truth detection.
>
>     Regarding the concern about the coupling of detection and recognition, we emphasize a key distinction discussed in Lines 658-660 of our paper. For specialized text spotters, recognition is strictly coupled with detection (i.e., a detection failure inherently results in a recognition failure). In contrast, MLLM-based recognition does not strictly rely on accurate text detection steps. MLLMs can leverage global context to recognize text content even if the specific localization (bounding box) is less precise. This decoupling is supported by our results: while specialized models may achieve higher detection precision (Table 2), our model achieves superior end-to-end recognition performance (Table 1).
>
> 3. No, we do not eliminate duplicate words. Although Eq. 6 uses the term "set," our implementation treats predicted and ground truth text strings as **Multisets (Bag of Words)** to handle duplicate instances. Specifically, the intersection count for a given word $w$ is calculated as $\min(\text{count}(w, P_{words}), \text{count}(w, GT_{words}))$. This mechanism ensures that valid duplicates (e.g., "Sale" appearing twice in the image) must be predicted the correct number of times to achieve full recall. Conversely, if the model hallucinates duplicates (e.g., predicting "Together" twice when it appears only once), the redundant prediction contributes to the total prediction count $|P_{words}|$ but not the intersection, thereby correctly penalizing the precision score. We will clarify this "Multiset" definition in the revised paper.
>
> 4. We apologize for this oversight and will ensure Figure 2 is properly referred in the revised paper.
>
> 5. First, we candidly acknowledge that since our fine-tuning process exclusively utilized text spotting datasets, there is some minor degradation in performance on general metrics. This is a common challenge in LLM fine-tuning, often referred to as catastrophic forgetting.
>
>     However, as discussed in our paper, we employ LoRA fine-tuning, which freezes the vast majority of pre-trained parameters and updates only a small set of adapters. This design inherently mitigates the extent of forgetting by preserving the model's core visual understanding capabilities. Our training primarily aligns the model's output format and refines spatial reasoning for text rather than overwriting its fundamental world knowledge. To quantify this, we present the table below comparing the performance of our model before and after SupGRPO training on some VQA benchmarks.
>
>     | Benchmark | Baseline (Qwen2.5-VL-7B) | Ours (SupGRPO) |
>     | :--- | :--- | :--- |
>     | DocVQA | 95.7 | 94.0 |
>     | ChartQA| 87.3 | 85.1 |
>     | MMMU | 58.6 | 57.7 |

---

### Author Response · Authors · 2025-11-24

We sincerely thank all reviewers for their constructive comments and insightful suggestions. We have provided detailed, point-by-point responses to address the questions and concerns raised by each reviewer. We have also updated the revised paper, incorporating these valuable suggestions.

---

### Meta-Review · Area_Chair_sLdC · 2025-12-21

**Summary:**

This paper proposes SupGRPO, a method that combines Supervised Fine-Tuning (SFT) and Group Relative Policy Optimization (GRPO) for improving text spotting capabilities of multimodal large language models (MLLMs). The authors observe that SFT excels at detection while GRPO excels at recognition, and propose a joint training strategy that combines both methods. They introduce a matching-based online SFT mechanism applied only to coordinate tokens and design task-specific reward functions. Additionally, they curate the ATS dataset for artistic text spotting.

While the paper addresses a relevant problem and demonstrates empirical improvements, it suffers from limited technical novelty, experimental fairness issues in the original submission, and narrow scope. The core contribution is primarily an engineering combination of existing techniques with incremental modifications. The authors' rebuttal provides new experiments that address several concerns, but the fundamental issues regarding novelty and generalizability remain unresolved.

Moreover, after carefully reviewing both the original paper and the evaluation feedback, we have identified several critical concerns. First, the performance of the proposed method still falls behind specialized text-spotting models. For instance, on the Total-Text dataset, LRANet++ achieves 90.8 while this work reports 89.6 in the rebuttal; similarly, on ATS, ABINet++ reaches 88.2 compared to this work's 88.0. Second, we note an apparent anomaly in Table 1 regarding the TS-VL-3B (Qwen2.5-VL-3B) model's results on the Total-Text dataset: the performance under the “None” context setting (92.4) and “Full” context setting (90.3) is inconsistent with the general trend observed across all other data, raising concerns about the reliability and interpretation of these results.

**Reviewer Concerns:**

**Concerns Addressed by the Rebuttal:**

The authors provided comprehensive responses to most concerns:

*   **Experimental Fairness:** The authors clarified that all baseline models were fine-tuned on identical training data including ATS, and specialized models were also fine-tuned on ATS. This addresses the fairness concerns raised by p8Ee and SV9w.

*   **Missing Comparisons:** The authors added comparisons with OCR-optimized models (GOT-OCR 2.0, Ocean-OCR, TextMonkey), directly addressing concerns from p8Ee and SV9w. However, these models significantly underperform, which raises questions about whether they are appropriate baselines.

*   **Evaluation Clarity:** The authors clarified that recognition accuracy refers to end-to-end recognition (full-image text spotting) rather than recognition on cropped regions. They also explained the multiset handling for duplicate words.

*   **Two-Stage Training:** The authors provided new experiments comparing joint training (SupGRPO) with two-stage training (SFT followed by GRPO), showing joint training's superiority. This addresses Reviewer p8Ee's concern.

*   **Architectural Generalization:** The authors demonstrated results on three different model scales/architectures (Qwen2.5-VL-3B, Qwen2.5-VL-7B, Qwen3-VL-8B), addressing Reviewer p8Ee's concern about generalization.

*   **Ablation Study Inconsistency:** The authors clarified that the "Baseline" in Table 7 refers to the vanilla model without RL, and the comparison is between Text-only vs. combined rewards, resolving Reviewer SV9w's concern.

**Outstanding Concerns:**

*   **Limited Technical Novelty:** Despite the rebuttal, the fundamental concern about limited novelty remains. The core contribution is combining SFT and GRPO with a matching mechanism. While this shows good engineering, the technical novelty is incremental. The matching-based online SFT is the main novel component, but it is a relatively straightforward modification.

*   **Narrow Scope and Generalizability:** The method is specifically designed for text spotting and shows degradation on other tasks. The paper does not address whether the approach could generalize to other vision-language tasks or whether it is fundamentally limited to text spotting.

*   **Original Submission Quality:** The fact that critical experiments (two-stage training comparison, architectural generalization, OCR-optimized model comparisons, full fine-tuning results) were missing from the original submission raises concerns about the thoroughness of the initial work. These should have been included in the original submission, not added during rebuttal.

*   **Practical Impact:** While the paper shows improvements over the baseline MLLM, the practical advantage over specialized models remains marginal in many cases. The original results showed a significant gap, and even the new results show that specialized models remain competitive.

**Reviewer Scores:**

Only Reviewer SV9w had participated in the discussion and maintained his score.

---

### Decision · Program_Chairs · 2026-01-26

Reject